# Assessment of functional diversities in patients with Asthma, COPD, Asthma-COPD overlap, and Cystic Fibrosis (CF)

**Richard Kraemer**[1,2,3]*, **Florent Baty**[4], **Hans-Jürgen Smith**[5], **Stefan Minder**[1], **Sabina Gallati**[2,6], **Martin H. Brutsche**[4], **Heinrich Matthys**[7]

**1** Centre of Pulmonary Medicine, Hirslanden Hospital Group, Salem-Hospital, Bern, Switzerland,
**2** Department of Paediatrics, University of Bern, Bern, Switzerland, **3** School of Biomedical and Precision Engineering (SBPE), University of Bern, Bern, Switzerland, **4** Department of Pneumology, Cantonal Hospital St. Gallen, St. Gallen, Switzerland, **5** Medical Development, Research in Respiratory Diagnostics, Berlin, Germany, **6** Hirslanden Precise, Genomic Medicine, Hirslanden Hospital Group, Zollikon/Zürich, Switzerland, **7** Department of Pneumology, University Hospital of Freiburg, Freiburg, Germany

* richard.kraemer@hirslanden.ch

**Data Availability Statement:** All relevant data are within the manuscript and its Supporting information files.

## Abstract

The objectives of the present study were to evaluate the discriminating power of spirometric and plethysmographic lung function parameters to differenciate the diagnosis of asthma, ACO, COPD, and to define functional characteristics for more precise classification of obstructive lung diseases. From the databases of 4 centers, a total of 756 lung function tests (194 healthy subjects, 175 with asthma, 71 with ACO, 78 with COPD and 238 with CF) were collected, and gradients among combinations of target parameters from spirometry (forced expiratory volume one second: $FEV_1$; $FEV_1$/forced vital capacity: $FEV_1/FVC$; forced expiratory flow between 25–75% FVC: $FEF_{25-75}$), and plethysmography (effective, resistive airway resistance: $sR_{eff}$; aerodynamic work of breathing at rest: sWOB), separately for in- and expiration ($sR_{eff}^{IN}$, $sR_{eff}^{EX}$, $sWOB_{in}$, $sWOB_{ex}$) as well as static lung volumes (total lung capacity: TLC; functional residual capacity: $FRC_{pleth}$; residual volume: RV), the control of breathing (mouth occlusion pressure: $P_{0.1}$; mean inspiratory flow: $V_T/T_I$; the inspiratory to total time ratio: $T_I/T_{tot}$) and the inspiratory impedance ($Z_{in}^{pleth} = P_{0.1}/V_T/T_I$) were explored. Linear discriminant analyses (LDA) were applied to identify discriminant functions and classification rules using recursive partitioning decision trees. LDA showed a high classification accuracy (sensitivity and specificity > 90%) for healthy subjects, COPD and CF. The accuracy dropped for asthma (~70%) and even more for ACO (~60%). The decision tree revealed that $P_{0.1}$, $sR_{tot}$, and $V_T/T_I$ differentiate most between healthy and asthma (68.9%), COPD (82.1%), and CF (60.6%). Moreover, using $sWOB_{ex}$ and $Z_{in}^{pleth}$ ACO can be discriminated from asthma and COPD (60%). Thus, the functional complexity of obstructive lung diseases can be understood, if specific spirometric and plethysmographic parameters are used. Moreover, the newly described parameters of airway dynamics and the central control of breathing including $Z_{in}^{pleth}$ may well serve as promising functional marker in the field of precision medicine.

**Funding:** The author(s) received no specific funding for this work.

**Competing interests:** The authors have declared that no competing interests exist.

## Introduction

There is an ongoing and growing interest in characterizing functional diversities by functional traits within obstructive lung diseases such as asthma, different phenotypes of chronic obstructive pulmonary disease (COPD) and cystic fibrosis (CF) [1–11]. COPD is a common, complex and heterogeneous disease, characterized by airflow limitation and an increased inflammatory response of the lung [12]. Noteworthy, a substantial proportion of patients show characteristics of both, asthma and COPD, referred to as the asthma–COPD overlap (ACO) [13–22]. Although a significant individual heterogeneity within COPD is well-known reflecting divers clinical patterns by different physiological mechanisms, endo-types and phenotypes [23], it is yet not possible to predict morbidity and mortality from the degree of lung function impairment in COPD [4]. CF is a severe, monogenic, autosomal recessive disease, caused by mutations in the cystic fibrosis transmembrane conductance regulator (CFTR) gene, where disturbed chloride and bicarbonate transportation in epithelial cells results in a multiorgan disease with primarily pulmonary infections and pancreatic insufficiency [24]. New therapies with CFTR modulators have shifted the previously symptomatic treatment adjusted to the patients' phenotype toward a genotype-specific treatment in the sense of precision medicine [10, 24]. It is important to recognize, that these conditions have differentiating features related to etiology, symptoms, type of airway inflammation, inflammatory cells and mediators, consequences of inflammation, response to therapy, and disease course. Therefore, a more targeted and holistic management has been claimed [4].

Precision medicine as an approach for tailoring disease treatment and prevention, is hoped to be the future of asthma, COPD and CF, enabling sub-classification as diagnostic, prognostic, or predictive response characteristics [1–10]. To focus on the definition of functional traits, such as biomarkers [1, 6], changes in the extracellular matrix [7], imaging modalities, prediction rules, and genetic factors, has already been proposed. However, in COPD, functional parameters defining the pathophysiologic processes are mainly based on spirometric parameters, predominantly on $FEV_1$ [25]. Surprisingly, there are only a limited number of studies, especially also no clinical trials, integrating plethysmographic parameters, and there is a lack of comparative parameters in the assessment of obstructive lung diseases on a same level of extended lung function testing, how it was performed previously for monitoring chronic lung diseases [26–29]. Having already demonstrated the discriminative power of certain traits in COPD [30–33], and patients with CF [34, 35], we intended to search for the discriminating power of further parameters, obtained by the airway resistance ($sR_{aw}$) loop. Fig 1 shows a $sR_{aw}$-loop consisting of the plethysmographic shift volume ($V_{pleth}$) and the tidal flow (V') plot, obtained in a patient with COPD. Noteworthy, aerodynamic parameters, such as sWOB and $sR_{eff}$ could also be computed for the inspiratory and expiratory part of the breathing cycle separately giving parameters such as $sWOB_{in}$, $sWOB_{ex}$, $sR_{eff}^{IN}$, and $sR_{eff}^{EX}$.

Based on the plea for the use of independent discriminatory parameters by Lopez and Centanni [9], and the new concept of "artificial intelligence" proposed by Topalovic et al. [26], the present study was designed with the aim of highlighting and comparing the various functional factors, and the physiological complexity within and between asthma, ACO, COPD and CF, using an extended set of spirometric and plethysmographic parameters in a multivariate approach, thus enabling the identification of functional traits within these diagnosis of obstructive pulmonary diseases.

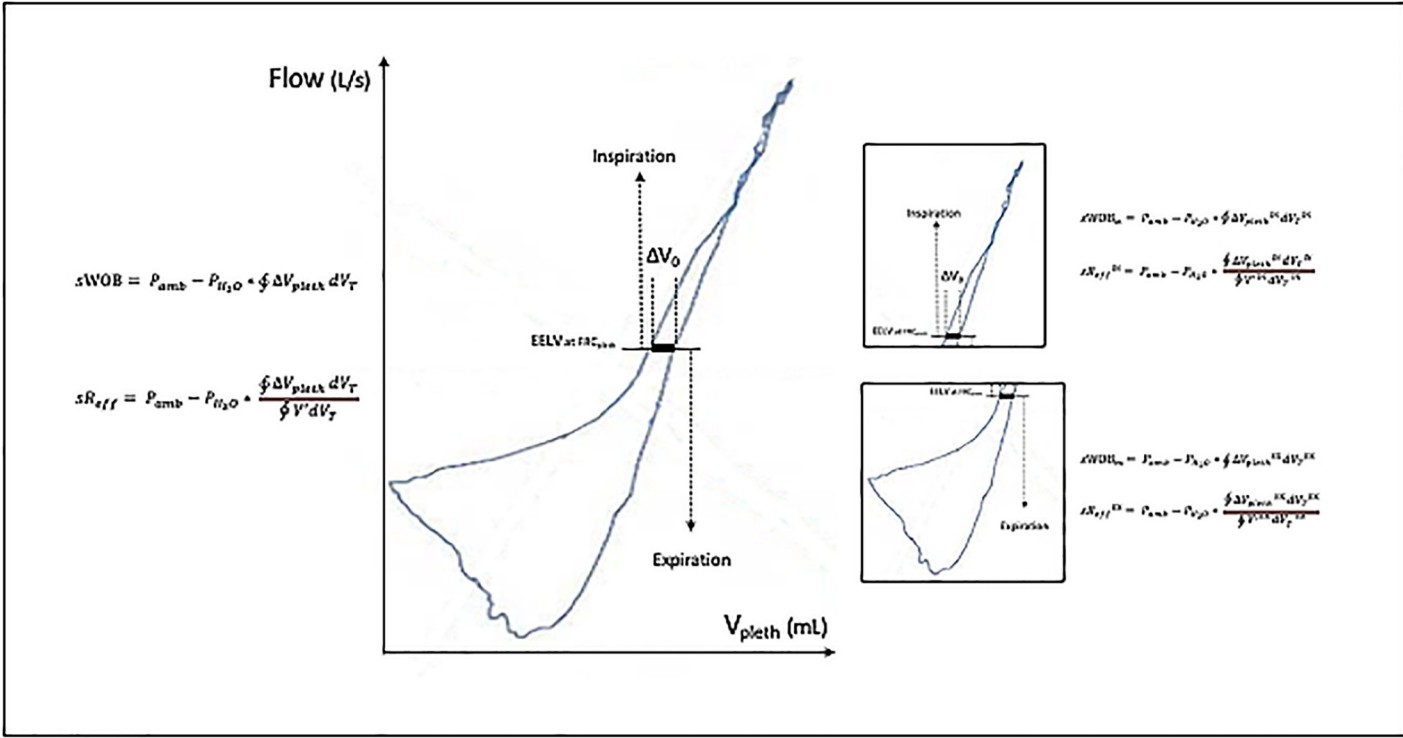

**Fig 1. Aerodynamic parameters computed by integrals from a plethysmographic shift volume—Tidal flow loop ($sR_{aw}$-loop) obtained from a patient with COPD, separated into the inspiratory and expiratory area of the loop.** ($V_{pleth}$: plethysmographic shift volume; EELV: end-expiratory lung volume. $FRC_{pleth}$: functional residual capacity; $\Delta V_0$: difference between inspiratory and expiratory shift-volume at $FRC_{pleth}$; sWOB: resistive aerodynamic work of breathing; $sR_{eff}$: effective specific airways resistance; $sWOB_{in}$: resistive aerodynamic work of breathing integrated from the inspiratory part of the $R_{aw}$-loop; $sWOB_{ex}$: resistive aerodynamic work of breathing integrated from the expiratory part of $sR_{aw}$-loop; $sR_{eff}^{IN}$: inspiratory, effective specific airways resistance; $sR_{eff}^{EX}$: expiratory, effective specific airways resistance).

## Material and methods

### Study design and ethics

In the present study, we refer on retrospectively evaluated data obtained from four Swiss centers (University Children's Hospital, Bern; Center of Pulmonary Diseases, Hirslanden Hospital Group, Salem-Hospital, Bern Switzerland; Clinic of Pneumology, Cantonal Hospital St. Gallen, Switzerland; Center of Pulmonology, Hirslanden Hospital Group, Clinic Hirslanden, Zürich, Switzerland), tested between 2006 and 2016. The patients have been referred to the centers for extended pulmonary function testing and optimizing therapy. Data were exported from the database systems of each clinic subdivided into 5 diagnostic classes: (*i*) healthy controls (*ii*) bronchial asthma, (iii) COPD, including a group of patients with (*iv*) COPD with coexisting asthma, (ACO), and (*v*) cystic fibrosis (CF), exported between 2018–2022. Authors had no access to information that could identify individual participants during or after data collection. The anamnestic, clinical features, and the diagnosis for each patient was made by trained pediatric and adult pulmonologists based on history-taking, chest radiographs, high-resolution CT scans, spirometry, whole-body plethysmography, and measurement of the fraction of exhaled nitric oxide (FeNO); additional detail regarding how the clinical diagnoses have been established previously given [32, 33].

The study was planned according to the Federal Law of Human Research, conceptualized according to the Swiss Ethics Committees on research involving humans, and approved by the

Governmental Ethic Committees of the States of Bern, St. Gallen and Zürich (Project KEK-BE PB_2017–00104). Master-files haven been stored and secured in the Clinical Trial Unit (CTU), Hirslanden, Corporate Office, CH-8152 Glattpark, Switzerland, and all relevant data are within the manuscript and its Supporting Information files.

## Patient's cohort and differential diagnosis

From the database of the four centers 756 measurement-sets of 194 healthy subjects, 175 with asthma, 71 with ACO, 78 with COPD and 238 with CF were collected. COPD was defined by a history of smoking (current or ex-tobacco smokers), or equivalent indoor/outdoor air pollution, with chronic cough, sputum and dyspnea, previously documented persistent airflow limitation with post-bronchodilator values of the $FEV_1/FVC < 70\%$ and the $FEV_1 < 80\%$, not fully reversible with a bronchodilator [36–38]. Asthma was diagnosed based on a past history of atopy and/or allergies with symptoms such as wheezing, shortness of breath, chest tightness and cough that vary over time in their occurrence, frequency, and intensity, and proven bronchial hyperreactivity (BHR) [13], or a positive bronchodilation test [39]. ACO was diagnosed when the subject had features of COPD and asthma, with documented bronchodilator response in a FEV1 > 12% and 200 mL [13, 21, 22, 40, 41]. Patients with CF were recruited from the Bernese Cystic Fibrosis Data Base [34, 42, 43], regularly seen at the outpatient clinic of the Department of Paediatrics. The diagnosis of CF was based on characteristic phenotypic features [44], confirmed by a duplicate quantitative pilocarpine iontophoresis sweat test measuring both Na and Cl values > 60 mEq/L, as well as by genotype identification using extended mutation screening [45, 46]. Additional regarding genotype analysis are given in the supporting information captions (S1: Genotype analysis in S1 File).

## Pulmonary function procedures

In all 4 centers the same type of a constant-volume whole-body plethysmographs (Master Screen Body, Jaeger Würzburg, Germany) were used by standard techniques according to ATS-ERS criteria [47–50] and revised Swiss guidelines [51]. The exported data were obtained from the same system software (JLAB, vers. 5.2, SentrySuite vers. 1.29 resp.). Inclusion criteria were reproducible base-line measurements with *a*) at least 5 shift volume-tidal flow loops of comparable shapes, *b*) especially closed at zero flow points, *c*) closed inspiratory part in the shift volume-tidal flow loops. All parameters were assessed in absolute values, as percentage of predicted normal values, and z-scores according to normative equations recently used [52, 53], and additionally given in the supporting information captions (S1: Pulmonary function procedures in S1 File). Apart from the extension of parameters obtained by the $sR_{aw}$-loop, we found it important to introduce also parameters defining the control of breathing. As initially worked out by Whitelaw et al. [54], the respiratory drive ($P_{0.1}$) was measured by means of a mouth occlusion pressure measurement 100 ms after inspiration as automatic occlusion response during tidal breathing. This makes the $P_{0.1}$ effort-independent, reproducible, and minimizes vagal influences because pressure swings do not lead to corresponding changes in volume [56]. Since it starts from end-expiratory lung volume (EELV), any drop in $P_{0.1}$ is independent of the recoil pressure of the lung or thorax and airway resistance because the flow is interrupted [55]. Moreover, effective inspiratory impedance defined as product of $P_{0.1}$ and the ratio between $V_T$ and the inspiratory time ($T_I$) was calculated [56, 57].

The content of the $sR_{aw}$-loop presented in the shift volume ($V_{pleth}$)—tidal flow (V')—plot (Fig 1) seems to be rather complex, especially in patients with COPD. The $sR_{aw}$-loop shows the typical pattern of a golf club in the expiratory limb, which is a sign of airflow limitation in the peripheral airways. The $sR_{eff}$-approximation of the $sR_{aw}$-loop and its reciprocal value, the

effective, resistive airway conductance ($sG_{eff}$), were proven to be target parameters reflecting small airways function [8, 26–28]. Details regarding the historical evolution how airway resistances can be calculated and hence computed are given in the supporting information captions (S2: Assessment of airway dynamics in S1 File). For all these parameters normative equations and values predicted could be computed [58]. They are given in details in the supporting information captions (S4: Defining predictive equations of airway dynamics in S1 File).

## Statistical approach and parameter modelling

The distribution of all parameters presented as age- and gender-corrected z-scores is reported. Two-sided tests with a type-I error α = 5% were used. *P*-values under 0.05 were considered as statistically significant. There were 5 diagnostic classes (healthy, asthma, ACO, COPD and CF) to be discriminated. Partly based on our former experience [33], parameters were grouped within categories including (*i*) static lung volumes (TLC, $FRC_{pleth}$, RV), (*ii*) breathing pattern ($V_T$, $V_T/FRC$), (*iii*) airway dynamics (sWOB, $sWOB_{in}$, $sWOB_{ex}$, $sR_{eff}$, $sR_{eff}^{IN}$, $sR_{eff}^{EX}$, $sR_{tot}$), (*iv*) forced spirometry ($FEV_1$, $FVC/FEV_1$, $FEF_{25-75}$), (*v*) control of breathing ($P_{0.1}$, $V_T/T_I$, $T_I/T_{tot}$), and (*vi*) effective inspiratory impedance ($Z_{in}^{pleth}$).

Principal component analyses (PCA) were used to explore the variance present in our data set. Linear discriminant analysis was performed to explore the inter-class variability, and functions discriminating between the 5 diagnoses were identified. A confusion matrix summarizing the classification accuracy of lung functions after leave-one-out cross-validation was created. Wilks's lambda (Λ) test statistics was used for refined variable selection. Conditional inference recursive partitioning trees were built and the importance of the different functional parameters across the diagnostic classes was estimated using a resampling-based performance procedure. All analyses were done using the IBM SPSS software (version 29.0; SPSS Inc., Chicago, IL), and the R statistical software, version 4.1.2 (R Statistics, Vienna, Austria), together with the extension packages MASS, FactoMineR, caret, rpart and ade4.

## Results

The anthropometric data of the healthy subjects and the patients within the 5 diagnostic classes are described in Table 1. There is a certain imbalance of the numbers, especially rather few COPD because of lacking parameters. Due to the circumstance CF is diagnosed shortly after birth [59], the mean age of CF patients was significantly lower compared with the other collectives.

### Assessment of functional deficits

As advocated recently, the use of individual z-scores was applied to assess severity instead of percentage predicted values, especially if lung function data within several diagnostic classes and different functional severities are assessed over a longer age range [60–62]. By that between-subject, age- and growth-related variability of the distribution of the reference population is considered (Table 1). Regarding potentially discriminating parameters between the diagnostic classes using the *F*-statistic of ANOVA, highest mean differences were obtained by $sWOB_{ex}$ (ACO vs. COPD: -22.5 SDS; p<0.001, COPD vs. CF: 21.3 SDS; p<0.001) followed by $Z_{in}^{pleth}$ (COPD vs. CF: -21.1SDS) and $sWOB_{in}$ (ACO vs. COPD: -18.3 SDS; p<0.001, COPD vs. CF: 15.4 SDS; p<0.001). Regarding distinction between Asthma and ACO high mean differences were found for $Z_{in}^{pleth}$ (9.2 SDS) and $sWOB_{ex}$ (8.6 SDS). $FEV_1$ expressed in % pred. presented with high mean differences between ACO and COPD (31.1 SDS; p<0.001) and between COPD and CF (-13.9 SDS). However, if presented in z-scores, the mean differences were not significant different (Table 1).

**Table 1. Anthropometric data within the collectives of the healthy subjects and patients with obstructive lung diseases, and group means of each lung function parameter expressed as z-scores.**

| | Healthy | Asthma | ACO | COPD | CF | All | Mean Diff. Asthma vs. ACO | Mean Diff. ACO vs. COPD | Mean Diff. COPD vs. CF |
|---|---|---|---|---|---|---|---|---|---|
| Measurements n (% total) | 194 (25.7) | 175 (23.1) | 71 (9.4) | 78 (10.3) | 238 (31.5) | 756 (100) | | | |
| Gender (male/female), n | 83/113 | 47/126 | 36/34 | 39/40 | 123/115 | 331/431 | | | |
| Age (mean in years) min / max | 39.3±20.1 5.9 / 85.7 | 42.3±20.1 6.7 / 83.6 | 55.4±18.0 18.2 / 87.2 | 69.8±9.8 38.4 / 92.2 | 13.4±5.6 5.1 / 31.4 | 36.6±24.0 5.1 / 92.2 | 13.0 p<0.001 | -15.1 p<0.001 | 56.8 p<0.001 |
| BMI (mean in kg/m$^2$) min / max | 23.4±4.1 12.9 / 29.9 | 25.1±4.8 14.4 / 38.3 | 25.5±4.2 19.2 / 37.7 | 26.1±5.6 14.3 / 45.7 | 17.2±2.2 12.2 / 24.3 | 22.3±5.4 12.2 / 45.7 | -.05 n.s. | -0.4 n.s. | 8.9 p < .001 |
| **Spirometry** | | | | | | | | | |
| $FEV_1$ z-score ±SD min / max | 0.0±0.9 -1.9 / 2.6 | -0.5±1.0 -2.9 / 2.4 | -1.0±1.2 -3.0 / 1.6 | -2.8±1.2 -4.7 / -0.3 | -2.6±1.2 -7.9 / 1.4 | -1.3±1.9 -7.89 / 2.7 | -0.6 p<0.05 | 1.7 p<0.001 | -0.2 n.s. |
| FEV1% pred ±SD min / max | 100.7±11.5 79.6 / 135.9 | 94.5±12.4 62.8 / 127.8 | 86.8±16.2 60.4 / 127.5 | 55.8±17.4 31.8 / 94.7 | 69.8±24.2 19.2 / 125.3 | 83.6±23.5 19.2 / 135.9 | 7.7 p<0.02 | 31.1 p<0.001 | -13.9 p<0.001 |
| $FEV_1/FVC$ z-score ±SD min / max | -0.6±0.1 -1.8 / 5.0 | -0.4±0.1 -4.0 / 2.3 | -1.3±0.1 -4.1 / 3.4 | -1.2±0.1 -7.4 / 0.6 | -1.2±0.1 -8.8 / 2.5 | -0.8±0.1 -8.8 / 9.0 | 0.4 n.s | 0.5 n.s. | -0.1 n.s. |
| $FEF_{25-75}$ z-score ±SD min / max | 0.3±0.1 -1.9 / 2.7 | -0.4±0.1 -2.4 / 1.7 | -.08±0.1 -3.2 / 0.4 | -1.5±0.1 -3.4 / 0.4 | -2.1±0.1 -5.2 / 1.4 | -0.9±0.1 -5.2 / 2.7 | 0.5 n.s. | 0.6 p<0.05 | 0.7 p<0.001 |
| **Airway Dynamics** | | | | | | | | | |
| sWOB z-score ±SD min / max | 0.0±0.1 -2.3 / 2.0 | 2.6±0.4 -2.8 / 18.8 | 7.1±0.5 -1.5 / 17.8 | 15.4±0.5 4.77 / 22.80 | 6.2±0.3 -2.4 / 17.5 | 4.81±0.2 -2.8 / 22.8 | -4.5 p<0.001 | -8.3 p<0.001 | 9.3 p<0.001 |
| $sWOB_{in}$ z-score ±SD min / max | -0.0±0.1 -2.1 / 2.3 | 2.4±0.4 -8.4 / 26.1 | 5.6±0.6 -9.4 / 18.7 | 23.0±0.9 4.3 / 41.9 | 8.5±0.5 -7.7 / 30.9 | 6.2±0.3 -9.4 / 41.9 | -3.2 p = 0.02 | -18.3 p<0.001 | 15.4 p<0.001 |
| $sWOB_{ex}$ z-score ±SD min / max | 0.0±0.1 -3.8 /2.2 | 6.8±8.7 -5.6 / 39.2 | 15.±7.6 0.3 / 34.3 | 37.9±9.5 15.1 /55.4 | 16.6±9.7 -2.0 /41.4 | 12.2±13.5 -5.6 / 55.4 | -8.6 p<0.001 | -22.5 p<0.001 | 21.3 p<0.001 |
| $sR_{eff}$ z-score ±SD min / max | -0.0±0.1 -1.9 / 2.0 | 3.3±0.3 -1.9 / 22.0 | 6.2±0.4 -1.7 / 14.6 | 15.7±0.6 3.3 / 24.6 | 10.4±0.3 -0.6 / 21.7 | 6.2±0.2 -1.9 / 24.6 | -3.0 p<0.001 | -9.5 p<0.001 | 5.3 p<0.001 |
| $sR_{eff}^{IN}$ z-score ±SD min / max | 0.1±0.1 -1.8 / 1.8 | 8.2±1.4 -17.5 / 62.2 | 24.7±2.5 -13.9 / 76.1 | 81.7±33.2 9.0 / 125.0 | 47.0±1.8 -18.1 / 115.3 | 27.4±1.2 -18.1 / 125.0 | -6.5 p<0.001 | -10.7 p<0.001 | 6.9 p<0.001 |
| $sR_{eff}^{EX}$ z-score ±SD min / max | 0.01±0.01 -1.9 / 2.0 | 2.0±3.6 -3.7 / 14.3 | 3.8±4.4 -3.0 / 16.5 | 16.0±6.6 1.1 / 25.7 | 11.6±5.9 -1.9 / 26.0 | 6.1±7.3 -3.7 / 26.0 | -1.8 P = 0.05 | -12.2 p<0.001 | 4.3 p<0.001 |
| $sR_{tot}$ z-score ±SD min / max | 0.0±0.9 -1.7 / 1.7 | 3.0±2.5 -2.5 / 13.5 | 4.8±3.0 -2.3 /13.5 | 15.4±3.7 5.8 / 22.5 | 8.5±4.1 -1.9 / 20.1 | 5.4±5.6 -2.5 / 22.5 | -1.8 p<0.001 | -10.7 p<0.001 | 6.9 p<0.001 |
| **Control of breathing** | | | | | | | | | |
| $P_{0.1}$ z-score ±SD min / max | -0.0±0.9 -2.0 / 2.1 | 3.1±6.7 -8.9 / 22.3 | 10.5±9.1 -8.2 / 28.4 | 12.2±8.8 -7.4 / 26.4 | 18.1±5.7 6.2 / 28.6 | 8.6±9.6 -8.9 / 28.6 | -7.42 p<0.001 | -1.75 p<0.001 | -5.78 p<0.001 |
| $V_T/T_I$ z-score ±SD min / max | -0.1±0.9 -1.9 / 1.8 | 1.3±3.4 -5.7 / 11.3 | 3.6±4.8 -3.8 / 15.5 | 3.5±3.9 -3.4 / 12.1 | -1.3±2.9 -6.5 / 15.5 | 0.6±3.4 -6.5 / 15.5 | -2.25 n.s | 0.11 n.s. | 4.70 p<0.001 |
| **Inspir. Impedance** | | | | | | | | | |
| $Z_{in}^{pleth}$ z-score ±SD min / max | 0.0±1.0 -1.8 / 2.0 | 3.9±9.8 -8.9 / 45.0 | 13.2±13.2 -10.2 / 45.1 | 16.8±14.7 -9.9 / 49.1 | 37.8±20.8 3.4 / 11.2 | 15.8±21.1 -10.2 / 11.2 | -9.2 p<0.001 | -3.6 n.s. | -21.1 p<0.001 |
| **End-expiratory Level** | | | | | | | | | |
| $FRC_{pleth}$ z-score ±SD min / max | 0.5±0.5 -.8 / 1.8 | 0.6±0.9 -1.5 / 3.1 | 1.1±1.0 -1.0 / 3.8 | 2.9±1.2 0.0 /5.0 | 1.9±1.5 -1.8 / 6.4 | 1.3±1.4 -1.8 / 6.4 | -0.5 p<0.02 | -1.8 p<0.001 | 1.0 p<0.001 |

## Linear discriminant analysis (LDA)

Parameters specified into 5 categories including (*i*) static lung volumes (TLC, $FRC_{pleth}$, RV,), (*ii*) breathing pattern (RR, $V_T$, $V_T/FRC$), (*iii*) airway dynamics (sWOB, $sWOB_{in}$, $sWOB_{ex}$, $sR_{eff}$, $sR_{eff}^{IN}$, $sR_{eff}^{EX}$, $sR_{tot}$), (*iv*) forced spirometry ($FEV_1$, $FVC/FEV_1$, $FEF_{25-75}$), and (*v*) control of breathing and inspiratory impedance ($P_{0.1}$, $V_T/T_I$, $T_I/T_{tot}$, $Z_{in}^{pleth}$) were used to perform a LDA. Table 2 shows that there was a very high classification accuracy (sensitivity and

**Table 2. Classification accuracy for healthy, asthma, ACO, COPD and CF according to a linear discriminant analysis (LDA) based on 16 lung function parameters.**

|  | Healthy | Asthma | ACO | COPD | CF |
|---|---|---|---|---|---|
| Sensitivity | 0.9691 | 0.7086 | 0.5916 | 0.9359 | 0.9580 |
| Specificity | 0.9484 | 0.9398 | 0.9635 | 1.0000 | 0.9768 |
| Pos. pred. value | 0.8664 | 0.7799 | 0.6269 | 1.0000 | 0.9500 |
| Neg. pred. value | 0.9889 | 0.9146 | 0.9579 | 0.9927 | 0.9806 |
| Prevalence | 0.2566 | 0.2315 | 0.0939 | 0.1032 | 0.3148 |
| Balanced Accuracy | 0.9578 | 0.8242 | 0.7775 | 0.9680 | 0.9674 |

Accuracy: 0.87; 95% CI (0.84, 089); Kappa: 0.82

specificity > 95%) for healthy, COPD and CF. The sensitivity was lower in patients with Asthma (ca. 70%) and even lower in ACO (ca. 60%). ACO was the most difficult diagnostic category to be classified. A linear discriminant analysis based on all the 16 parameters is graphically represented in Fig 2, showing that the first function discriminates between healthy and CF, whereas the second function depicts a gradient discriminating gradually healthy, asthma, ACO and COPD. The overall prediction accuracy was 87% (healthy: 96%, asthma: 82%, ACO: 78%, COPD: 97% and CF: 97%). Comparing COPD with CF, mean differences were: $P_{0.1}$ lower (-5.78 SDS; p < .001), $V_T/T_I$ higher (4.70 SDS; p < .001), inspiratory impedance ($Z_{in}^{pleth} = P_{0.1}/V_T/T_I$) lower (-21.07 SDS; p < .001), $FRC_{pleth}$ higher (.97 SDS, p < .001), and $FEF_{25-75}$ lower (-.68 SDS; p < .001).

The most striking parameters differentiating between these diagnostic classes are shown in Fig 2. Using all 16 lung function parameters the 5 diagnostic classes could be differentiated with an overall prediction accuracy of 86% (healthy: 97%, asthma: 65%, ACO: 70%, COPD: 95% and CF: 93%). Fig 2 demonstrates in the discriminant analysis a most pronounced difference between COPD and CF. Based on Wilks's lambda ($\Lambda$) test statistics 5 parameters $sR_{tot}$ ($\Lambda$ = 0.332), $FEF_{25-75}$ ($\Lambda$ 0.321), $sR_{eff}^{EX}$ ($\Lambda$ = 0.313) $sWOB_{ex}$ ($\Lambda$ = 0.302), and $P_{0.1}$ ($\Lambda$ = 275) discriminated between COPD and CF in 96, 98% respectively. On the other hand, ACO which is the most discussed subtype of COPD could be discriminated against asthma and COPD with a group membership prediction of 76%.

## Conditional inference tree

A conditional interference tree was built to investigate discriminating lung function parameters and defined classification rules to differentiate between healthy subjects, patients with asthma, ACO, COPD, and CF. The recursive partitioning trees are shown in Fig 3. Independent parameters $P_{0.1}$, $sR_{tot}$, $V_T/T_I$ $FEV_1/FVC$, $P_{0.1}$ and $sR_{eff}^{EX}$ expressed in z-scores were selected. In Fig 3A decision tree was built with all subjects. $P_{0.1}$ was the most significant parameter stratifying the 756 measurements within the 5 subject groups. If $P_{0.1}$ was $\leq$ 6.15 and $sR_{tot}$ > 1.69 asthmatics could be differentiated from healthy subjects in 68.9% (Node 4), 79.5% resp. (Node 3). If $P_{0.1}$ was > 6.15 and $V_T/T_I \leq$ 0.92 CF could be differentiated in 82.6% (Node 5) from the group of COPD phenotypes. If apart from $P_{0.1}$ > 6.15 and $V_T/T_I \leq$ 0.92 $sR_{tot}$ > 10.28 COPD could be distinguished from ACO, however only in 36.8% (node 7). If admittedly for the decision between asthma, ACO and COPD $sWOB_{ex}$ and $Z_{in}^{pleth}$ are used as shown in Fig 3B, then ACO can be distinguished in 60% from asthma (node 5). The decision tree reveals that $P_{0.1}$, $sRt_{ot}$, $V_T/T_I$, $sWOB_{ex}$ and $Z_{in}^{pleth}$ presented as significant functional traits differentiating between these different obstructive lung diseases.

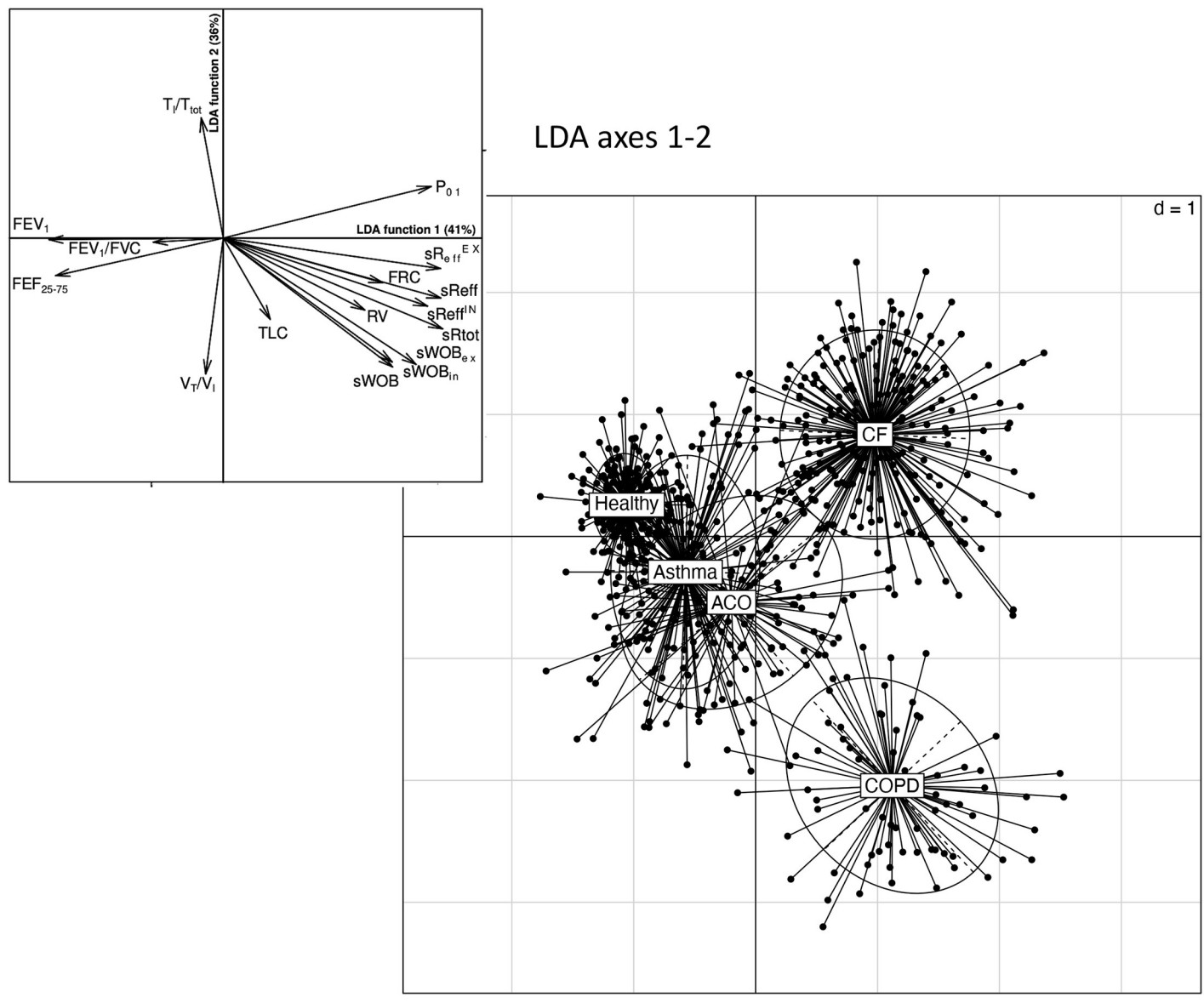

**Fig 2. Linear discriminant analysis (LDA): First function discriminates between healthy and CF, whereas the second function depicts a gradient discriminating gradually healthy, asthma, ACO and COPD, based on 16 lung function parameters selected by MANOVA.**

### Parameter-rating across the diagnostic classes

The variable importance was calculated based on a random cross-validation procedure following the conditional inference tree analysis. Table 3 shows the importance of the different functional parameters across the diagnostic classes. There were 16 lung function parameters which prompted as potential discriminators. The central respiratory drive $P_{0.1}$, the mean inspiratory flow $V_T/T_I$, the plethysmographic $sR_{eff}^{EX}$ and $sR_{tot}$, as well as the $sWOB_{ex}$ qualify as the 5 best discriminating determinants of the functional pattern of the four diseases studied. The best rated spirometric parameter was $FEF_{25-75}$ on place 7, followed by $FEV_1$ on place 10 and the $FEV_1/FVC$ ratio on place 14.

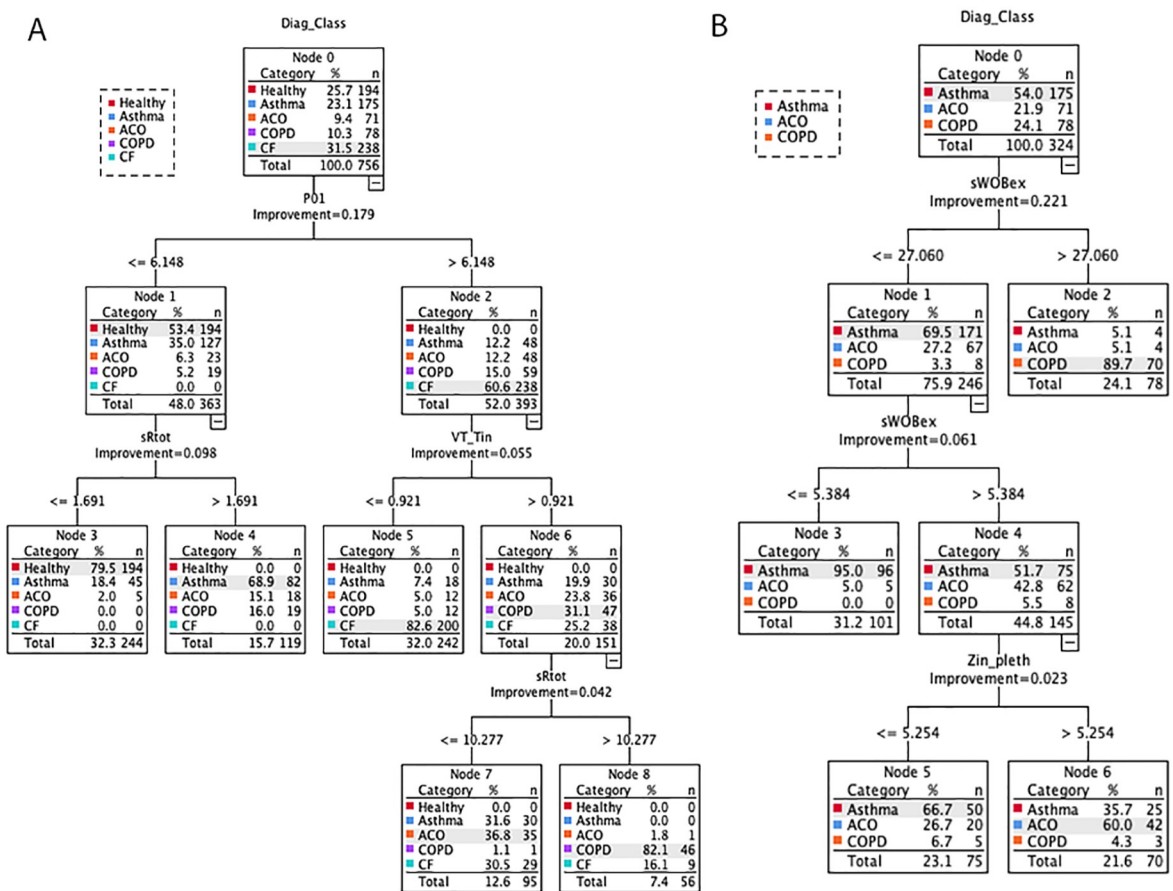

**Fig 3.** Decision-trees differentiating between healthy, asthma, ACO, COPD, CF involving all 16 lung function parameters, (left-hand part A) and differentiating asthma, ACO and COPD involving $sWOB_{ex}$, and $Z_{in}^{pleth}$ (right-hand part B).

**Table 3. Parameter-rating across the 4 diagnostic groups of 16 lung function parameters prompting as potential discriminators.**

| | Parameter | Asthma | ACO | COPD | CF |
|---|---|---|---|---|---|
| 1 | $P_{0.1}$ | 79.8 | 24.3 | 32.2 | 100.0 |
| 2 | $V_T/T_I$ | 23.8 | 43.4 | 34.7 | 62.0 |
| 3 | $sR_{eff}^{EX}$ | 36.5 | 19.5 | 29.4 | 61.1 |
| 4 | $sR_{tot}$ | 42.2 | 33.5 | 53.0 | 33.8 |
| 5 | $sWOB_{ex}$ | 37.4 | 24.0 | 49.6 | 33.2 |
| 6 | $sWOB$ | 39.2 | 33.6 | 42.1 | 48.4 |
| 7 | $FEF_{25-75}$ | 28.9 | 9.0 | 30.3 | 47.3 |
| 8 | $sWOB_{in}$ | 31.2 | 29.2 | 45.5 | 19.3 |
| 9 | $T_I/T_{tot}$ | 10.3 | 14.4 | 42.8 | 30.4 |
| 10 | $FEV_1$ | 42.3 | 12.2 | 24.7 | 38.6 |
| 11 | $FRC_{pleth}$ | 41.2 | 8.4 | 19.2 | 19.7 |
| 12 | $sR_{eff}$ | 40.1 | 23.8 | 34.6 | 40.1 |
| 13 | $sR_{eff}^{IN}$ | 38.3 | 7.0 | 31.9 | 31.2 |
| 14 | $FEV_1/FVC$ | 20.8 | 7.5 | 14.8 | 30.7 |
| 15 | $RV$ | 20.8 | 15.2 | 24.3 | 10.3 |
| 16 | $TLC$ | 13.0 | 20.1 | 22.2 | 14.9 |

## Discussion

Although COP, ACO and CF have some phenotypically associated clinical features such as chronic airway inflammation, recurrent infectious exacerbations, mucus hypersecretion, and impaired mucociliary clearance, they also share some key functional features such as progressive airflow obstruction, pulmonary hyperinflation, trapped gases and gas exchange disturbances, and hence that may suggest the existence of common mechanisms [63–66]. All three diseases have always been treated as unrelated, distinct entities. However, there is sparse knowledge regarding similarities or specific functional patterns distinguishing these diseases.

Functional hallmark of all the four diseases, asthma, ACO, COPD and CF is the airway obstruction. The parameters of airway dynamics obtained by integration of the plethysmographic $sR_{aw}$-loop measured during tidal breathing featuring sWOB and $sR_{eff}$, have already shown specific predictive power regarding bronchodilator response [32], onset and progression of disease in patients with sub-phenotypes of COPD [30–33], and in patients with CF [42, 43]. The present study, however, demonstrates, that further information can be gained from this $sR_{aw}$-loop, if the expiratory area is selectively evaluated from the inspiratory area.

### Findings of the present study

The main findings of the present study are that apart from the central respiratory drive $P_{0.1}$ and $sR_{tot}$, parameters such as $sWOB_{ex}$ and $sReff^{EX}$ qualify as discriminating determinants of the functional pattern of the four diseases studied (Table 2). It implies that these parameters may feature important trajectories for differentiating specific functional patterns of obstructive lung diseases, and could be introduced as treatable traits in future concepts of "artificial intelligence" [3–5, 11, 67]. In this way, subclassification could be strengthened as diagnostic, prognostic, or predictive response characteristics toward precision medicine for patients with obstructive lung diseases.

### Peripheral airway dysfunction

Mahut et al. suggested that $sR_{tot}$ and $sR_{eff}$ can be considered as equivalent and correlated with indices that are considered to explore peripheral airways, and that these two parameters are statistically linked to activity-related dyspnea in COPD [31]. The open shape of the $sR_{aw}$-loop is related to changes in the elastic recoil pressure and collapse during quiet tidal breathing due to the changes in airway obstruction visible between inspiratory and expiratory flow. The shape also indicates unequal ventilation of the lung areas, which is a typical early sign of a pulmonary disease. Moreover, we have previously shown that the area of the $sR_{aw}$-loop is related to the flow-resistive work of breathing sWOB [32], which may explain its correlation with activity-related dyspnea, as demonstrated by Mahut et al. [31]. The parameter $sR_{eff}$ is an outstanding descriptor of lung function as it incorporates airways resistance and volume components that may be related to the viscoelastic behavior of the lung. In patients with COPD, inhomogeneity of ventilation within the small airways causes the opening of the $sR_{aw}$-loop due to expiratory flow limitation and/or dynamic airway compression, being clearly linked to activity-related dyspnea [68]. The same phenomenon is also found in patients with CF. Therefore, $sR_{eff}$ and $sR_{tot}$ represent surrogates of activity-related dyspnea in moderate to severe obstructive pulmonary diseases. The present study demonstrates that beyond sWOB and $sR_{eff}$ much more information can be found if the integrated data of the $sR_{aw}$-loop are analyzed for the inspiratory and expiratory limb separately. Especially $sWOB_{ex}$ and $sReff^{EX}$ are important discriminative lung function parameters. The disposition of normative reference equations transitional applied over a wide age range are prerequisites for studies of predicting disease

progression in asthma, subtypes of chronic obstructive pulmonary diseases (COPD) and cystic fibrosis (CF).

There is growing interest in recognizing specific functional patterns by standardized interpretation of pulmonary function tests in the diagnosis of respiratory diseases, built on expert opinions within a concept of precision medicine [1, 4, 10, 23, 25, 67]. Recently, Topalovic et al. reported artificial intelligence-based software significantly improving clinical practice for powerful decisions to distinguish different respiratory diseases [67]. Apart from parameters representing inspiratory and expiratory parts of the $sR_{aw}$-loop, we thought it worthwhile to include parameters of the control of breathing ($P_{0.1}$, $V_T/T_I$, and $T_I/T_{tot}$) in such a multivariate discriminating model. Moreover, the airway dynamics parameters were specifically split for those representing the inspiratory and expiratory parts of the $sR_{aw}$-loop. Depending on which parameter set is used, the four diagnoses can be distinguished fro healthy subjects with an overall prediction accuracy of 86%. Most pronounced difference was found between COPD and CF with a prediction accuracy of 99.7%. Finally, ACO, the most discussed subtype of COPD, was distinguished from asthma and COPD with a prediction accuracy of 76%.

## Limitations and strengths of the study

Several limitations need to be mentioned: The present study is a retrospective evaluation of lung function data obtained by various parameters and there are no longitudinal observations, a feature that can only be achieved by a prospectively designed study. Our actual challenge was to find surrogate markers superior to the convential spirometric parameters, significantly helping in the differentiation between COPD and CF. As it turned out, such an evaluation based on various functional parameters of extended airway dynamics revealed a variety of fundamentally different archetypes.

The main limitation of our study is, that it addresses functional trajectories representing the complex lung physiology in COPD and CF, not directly linked to clinical settings. However, the aerodynamic specific work of breathing at rest (sWOB) features presumably the closest parameter associated with clinical signs such as wheezing, shortness of breath, chest tightness and cough, and could well be taken as a marker for longitudinal follow-up and treatment efficacy. Other limitations are the relatively small number of subjects per center and within the sub-groups of COPD. However, there were no differences when the data of the patients with asthma, ACO and COPD of the 3 centers (LZB, LZH, KSSG) were compared to one another given in the supporting information captions (S5: S1 Table in S1 File). Therefore, the population-based retrospective nature of our study and its highly standardized multicenter framework has reliable power.

The strengths of the present study are that we enlarged the possibilities of plethysmographic target parameters, differentiating between parameters obtained from the inspiratory versus expiratory part of the $sR_{aw}$-loop, allowing to examine the interrelationships between several facets of lung function trajectory within these diseases. In addition, our model was flexible in that it allowed changes in lung function, expressed as z-scores, to vary over time between different classes of lung function trajectories.

## Conclusions

Given the functional, structural, and biological heterogeneity in patients with subtypes of COPD and CF, we anticipate, that there is considerable interest in assessing the differences between these diseases, using a set of functional target parameters. By such an approach flow limitation, airway dynamics, small airways dysfunction and the control of breathing can interactively be evaluated, assessing specifically the complex diagnostic-class-specific functional

deficits. The summary of the various functional defects, their combination and their interactions underscore the heterogeneous physiological mechanisms of these diseases. Extended lung function testing could presumably help in tracking dynamics and changes over time in view of specific disease burden, and a more sophisticated assessment of functional deficits and their reversibility would be justified. There are unquestionably several subtypes of COPD that are clinically distinct from those of CF. Moreover, it has clearly been shown that the prognosis in CF largely depends on the patients' genotype, and hence on the residual function of CFTR. Since there are significant differences not only between COPD and CF, but also between the different subtypes, the clinical behavior of patients varies, requiring the definition of a differentiated treatment strategy in terms of precision.

## Supporting information

**S1 File.**
(DOCX)

## Acknowledgments

We thank the technicians of each center who volunteered for lung function testing diligently devoting their time. We also thank Mr. Sven Fassbinden for the support exporting the lung function data in each center.

## Author Contributions

**Conceptualization:** Richard Kraemer.

**Data curation:** Stefan Minder, Martin H. Brutsche.

**Formal analysis:** Richard Kraemer, Florent Baty, Sabina Gallati.

**Methodology:** Richard Kraemer, Hans-Jürgen Smith, Sabina Gallati.

**Supervision:** Florent Baty.

**Writing – original draft:** Richard Kraemer.

**Writing – review & editing:** Richard Kraemer, Florent Baty, Hans-Jürgen Smith, Stefan Minder, Sabina Gallati, Martin H. Brutsche, Heinrich Matthys.

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
