## [Decision Letter · Decision Letter 0]

10 Aug 2023

PONE-D-23-12620Assessment of functional diversities in patients with Asthma, Asthma-COPD overlap, and Cystic Fibrosis (CF)PLOS ONE

Dear Dr. Kraemer,

Thank you for submitting your manuscript to PLOS ONE. After careful consideration, we feel that it has merit but does not fully meet PLOS ONE’s publication criteria as it currently stands. Therefore, we invite you to submit a revised version of the manuscript that addresses the points raised during the review process. Please submit your revised manuscript by Sep 24 2023 11:59PM. If you will need more time than this to complete your revisions, please reply to this message or contact the journal office at plosone@plos.org. Please include the following items when submitting your revised manuscript:A rebuttal letter that responds to each point raised by the academic editor and reviewer(s). You should upload this letter as a separate file labeled 'Response to Reviewers'.A marked-up copy of your manuscript that highlights changes made to the original version. You should upload this as a separate file labeled 'Revised Manuscript with Track Changes'.An unmarked version of your revised paper without tracked changes. You should upload this as a separate file labeled 'Manuscript'.

We look forward to receiving your revised manuscript.

Kind regards,

Ruud AW Veldhuizen

Academic Editor

PLOS ONE

Journal Requirements:

- https://www.dovepress.com/front_end/bronchodilator-response-in-patients-with-copd-asthma-copd-overlap-aco--peer-reviewed-fulltext-article-COPD

- https://onlinelibrary.wiley.com/doi/10.1111/apm.12915

In your revision ensure you cite all your sources (including your own works), and quote or rephrase any duplicated text outside the methods section. Further consideration is dependent on these concerns being addressed.

6. Please remove your figure 1 from within your manuscript file, leaving only the individual TIFF/EPS image files, uploaded separately. These will be automatically included in the reviewers’ PDF.

7. Please upload a new copy of Figure 3 as the detail is not clear. Please follow the link for more information:

https://blogs.plos.org/plos/2019/06/looking-good-tips-for-creating-your-plos-figures-graphics/

https://blogs.plos.org/plos/2019/06/looking-good-tips-for-creating-your-plos-figures-graphics/

**Additional Editor Comments:**

As you can note from the reviewer's comments, there is a substantial discrepancy between the assessments of the two reviewers. In revising your manuscript please pay particular attention to the discussion of the manuscript and how the results are interpreted. In addition, please make sure clear definitions of the assessed groups are provided and discussed.   On a separate note, I like to apologize for the long time that has elapsed between your submission and providing you with these reviews.   

Reviewers' comments:

Reviewer's Responses to Questions

**Comments to the Author**

1. Is the manuscript technically sound, and do the data support the conclusions?

Reviewer #1: Yes

Reviewer #2: No

2. Has the statistical analysis been performed appropriately and rigorously? 

Reviewer #1: Yes

Reviewer #2: Yes

3. Have the authors made all data underlying the findings in their manuscript fully available?

Reviewer #1: Yes

Reviewer #2: No

4. Is the manuscript presented in an intelligible fashion and written in standard English?

Reviewer #1: Yes

Reviewer #2: Yes

5. Review Comments to the Author

Reviewer #1: 1. Abstract: several abbreviations used without definition.

2. All manuscript: There is main focus on COPD and CF; ACO and asthma are not mentioned in an adequate manner, consider the following, with a focus on phenotypes:

Alsayed, A. R., Abu-Samak, M. S., & Alkhatib, M. (2023). Asthma-COPD Overlap in Clinical Practice (ACO_CP 2023): Toward Precision Medicine. Journal of Personalized Medicine, 13(4), 677.

Adrish, M., Anand, M. P., & Hanania, N. A. (2022). Phenotypes of Asthma–Chronic Obstructive Pulmonary Disease Overlap. Immunology and Allergy Clinics, 42(3), 645-655.

3. The discussion part can be improved

Reviewer #2: I appreciated the opportunity to review " Assessment of functional diversities in patients with Asthma, Asthma-COPD overlap, and Cystic Fibrosis (CF)” by Richard Kraemer et al. Authors investigated various lung function parameters to discriminate airway disease. However, there are several concerns, mainly regarding concept of the study.

Main concern.

1. First, the purpose and results of the study do not match. In the discussion (Line 441 – 459), the authors mainly refer to the comparison and difference between COPD and cystic fibrosis. Cystic fibrosis is known as a risk factor for COPD, especially in young patients, so I wonder why it is necessary to evaluate the difference between cystic fibrosis and COPD rather than age-matched comparisons if authors wanted to know physiologic and functional differences between two groups. I think the main aim of the authors' research and the direction of interpretation and discussion of the results are different.

2. Second, a definition of airway disease must be presented. In particular, ACO is only mentioned as a subtype of COPD, but ACO itself is also a disease showing heterogeneity, so it is a patient group that needs to be classified through an appropriate definition.

3. Table 1 presents the demographic comparison between each group. Age raises a question. Is it correct that 14.3 and 18.2 year olds are COPD and ACO patients? Of course, the definition of COPD has been recently updated in GOLD, but it is necessary to specify what definition these patients were classified as COPD and ACO in their teens.

4. Regarding the implications of each lung function and plethymography parameter in airway disease, I feel that there is a lack of consideration on the meaning of indicators showing differences in asthma, ACO, COPD, and CF.

Minor comments.

1) Introduction Line 149.

In COPD, it is difficult to predict the prognosis only with the decline in lung function, so various prognostic risk factors for symptoms such as dyspnea, aggravation, and accompanying diseases have already been suggested. It would be better to present the analysis results on these data.

2) Introduction Line 176 & Discussion line 471

When the term ACO was first introduced in the introduction, it was presented only as an abbreviation. Conversely, the discussion explains the abbreviation of ACO. Edit is needed

3) Line 283, Please spell out the first abbreviation. For example SDS.

4) Discussion, Line 359-367

The description of COPD and CF is the same as the introduction because it is very redundant.

5) Line 490-493

It is thought that there is a limitation of generalization with a small number of retrospective studies.

6) Line 494-502

The authors explain the research results as a physiologic biomarker that is more reliable than existing lung function parameters, but there are limitations in suggesting differences and clinical implication among diseases with appropriate cutoff values. The fact that it is a value that can be applied flexibly is felt as a disadvantage rather than an advantage.

6. PLOS authors have the option to publish the peer review history of their article (what does this mean?). If published, this will include your full peer review and any attached files.

Reviewer #1: No

Reviewer #2: No

---

## [Author Response · Author response to Decision Letter 0]

10 Sep 2023

Review Comments to the Author

Reviewer #1: 

1. Abstract: several abbreviations used without definition.

RK.: done

2. All manuscript: There is main focus on COPD and CF; ACO and asthma are not mentioned in an adequate manner, consider the following, with a focus on phenotypes:

Alsayed, A. R., Abu-Samak, M. S., & Alkhatib, M. (2023). Asthma-COPD Overlap in Clinical Practice (ACO_CP 2023): Toward Precision Medicine. Journal of Personalized Medicine, 13(4), 677.

Adrish, M., Anand, M. P., & Hanania, N. A. (2022). Phenotypes of Asthma–Chronic Obstructive Pulmonary Disease Overlap. Immunology and Allergy Clinics, 42(3), 645-655.

RK.: We appreciate this statement and added the definition of ACO (lines 152-154), referencing the two proposed papers.

3. The discussion part can be improved

RK.: According to the criticism of both Reviewers and the Academic Editors, the discussion part was rigorously improved.

Reviewer #2: 

I appreciated the opportunity to review " Assessment of functional diversities in patients with Asthma, Asthma-COPD overlap, and Cystic Fibrosis (CF)” by Richard Kraemer et al. Authors investigated various lung function parameters to discriminate airway disease. However, there are several concerns, mainly regarding concept of the study.

Main concern.

1. First, the purpose and results of the study do not match. In the discussion (Line 441 – 459), the authors mainly refer to the comparison and difference between COPD and cystic fibrosis. Cystic fibrosis is known as a risk factor for COPD, especially in young patients, so I wonder why it is necessary to evaluate the difference between cystic fibrosis and COPD rather than age-matched comparisons if authors wanted to know physiologic and functional differences between two groups. I think the main aim of the authors' research and the direction of interpretation and discussion of the results are different.

RK.: We have difficulties to answer to this statement. By now means CF can be considered only as risk factor for COPD. This Reviewer is certainly aware of the fact, that these two diseases are completely different identities. An age-matched control is rarely possible, because the age-distributions of these 2 diseases are completely different. It was, however, thought-provoking for us, to search for the functional patterns within these two diseases.

2. Second, a definition of airway disease must be presented. In particular, ACO is only mentioned as a subtype of COPD, but ACO itself is also a disease showing heterogeneity, so it is a patient group that needs to be classified through an appropriate definition.

RK.: We appreciate this statement and added the definition of ACO (lines 152-154), referencing the two papers proposed by Reviewer 1.

3. Table 1 presents the demographic comparison between each group. Age raises a question. Is it correct that 14.3 and 18.2 year olds are COPD and ACO patients? 

RK.: There must be a mis-understanding: The patient aged 18.2 years was considered as ACO, because she presented with both, airway obstruction (sRtot: 1.9 SDS), increased work of breathing (sWOB 2.6 SDS, sWOBex: 7.2 SDS) and a DLCO of .1.77 SDS. She was considered as potentially developing ACO. There is now patient with age 14.3 classified as ACO or COPD.

Of course, the definition of COPD has been recently updated in GOLD, but it is necessary to specify what definition these patients were classified as COPD and ACO in their teens.

4. Regarding the implications of each lung function and plethymography parameter in airway disease, I feel that there is a lack of consideration on the meaning of indicators showing differences in asthma, ACO, COPD, and CF.

RK.: We appreciate this statement, and focus now more on the meaning of these indicators, especially the small airways dysfunction.

Minor comments.

5. Introduction Line 149.

In COPD, it is difficult to predict the prognosis only with the decline in lung function, so various prognostic risk factors for symptoms such as dyspnea, aggravation, and accompanying diseases have already been suggested. It would be better to present the analysis results on these data.

RK.: We appreciate this statement and corrected the wording of now Line 168-173 accordingly.

6. Introduction Line 176 & Discussion line 471

When the term ACO was first introduced in the introduction, it was presented only as an abbreviation. Conversely, the discussion explains the abbreviation of ACO. Edit is needed

RK.: corrected

7. Line 283, Please spell out the first abbreviation. For example SDS. 

RK.: corrected

8. Discussion, Line 359-367

The description of COPD and CF is the same as the introduction because it is very redundant. 

RK.: Thank you, this point is well taken. Wie rephrased the part accordingly

5) Line 490-493

It is thought that there is a limitation of generalization with a small number of retrospective studies. 

RK.: At least we found it necessary to mention this point

6) Line 494-502

The authors explain the research results as a physiologic biomarker that is more reliable than existing lung function parameters, but there are limitations in suggesting differences and clinical implication among diseases with appropriate cutoff values. The fact that it is a value that can be applied flexibly is felt as a disadvantage rather than an advantage.

RK.: We appreciate

PLOS authors have the option to publish the peer review history of their article (what does this mean?). If published, this will include your full peer review and any attached files. 

RK.: yes

Do you want your identity to be public for this peer review? For information about this choice, including consent withdrawal, please see our Privacy Policy.

Reviewer #1: No

Reviewer #2: No

---

## [Editor Report · Decision Letter 1]

18 Sep 2023

Assessment of functional diversities in patients with Asthma, Asthma-COPD overlap, and Cystic Fibrosis (CF)

PONE-D-23-12620R1

Dear Dr. Kraemer,

We’re pleased to inform you that your manuscript has been judged scientifically suitable for publication and will be formally accepted for publication once it meets all outstanding technical requirements.

Kind regards,

Ruud AW Veldhuizen

Academic Editor

PLOS ONE
---

## [Editor Report · Acceptance letter]

16 Oct 2023

PONE-D-23-12620R1 

Assessment of functional diversities in patients with Asthma, COPD, Asthma-COPD overlap, and Cystic Fibrosis (CF) 

Dear Dr. Kraemer:

I'm pleased to inform you that your manuscript has been deemed suitable for publication in PLOS ONE. Congratulations! Your manuscript is now with our production department. 

Kind regards, 

on behalf of

Dr. Ruud AW Veldhuizen 

%CORR_ED_EDITOR_ROLE%

PLOS ONE